# Exploiting Network Compressibility and Topology in Zero-Cost NAS

Lichuan Xiang[1*]  Rosco Hunter[1*]  Łukasz Dudziak[2*]  Minghao Xu[1]  Hongkai Wen[1,2]

[1]University of Warwick
[2]Samsung AI Center Cambridge, UK
[*]These authors contributed equally to this work

**Abstract**  Neural Architecture Search (NAS) has been widely used to discover high-performance neural network architectures over manually designed approaches. Despite their success, current NAS approaches often require extensive evaluation of candidate architectures in the search space, or the training of large super networks. To reduce the search cost, zero-cost proxies have recently been proposed as a way to efficiently predict the performance of an architecture. Though many novel proxies have been put forward in recent years, relatively little attention has been dedicated to pushing our understanding of the existing ones. Contrary to that trend, in our work, we argue that it is worth revisiting and analysing the existing proxies in order to further push the boundaries of zero-cost NAS. Towards that goal, we propose to view the existing proxies through a common lens of network compressibility, trainability, and expressivity. Notably, doing so allows us to build a better understanding of the high-level relationship between different proxies as well as refine some of them into their more informative variants. We leverage these insights to design a novel saliency and metric aggregation method informed by compressibility, orthogonality, and network topology. We show that our proposed methods are simple but powerful and yield state-of-the-art results across popular NAS benchmarks. Our code is available at: https://www.github.com/tiaspetto/T-CET

## 1 Introduction

The goal of Neural Architecture Search is to automatically identify high-performance neural network architectures with respect to some measure of success. NAS has three essential components (He et al., 2021): an appropriate search space of possible architectures; a search algorithm; and a metric that quantifies the performance of the architectures. The choice of metric determines the geometry of the optimisation landscape and, thus, the trajectory of the search algorithm. So, finding appropriate metrics is of foundational importance. Metrics that rapidly estimate architecture performance are used to reduce the search time and such metrics are often referred to as "zero-cost proxies" (White et al., 2021). There has been extensive work proposing novel metrics; however, means for improving existing metrics have received less attention. On one level, it is common practice for parameter-level information to be aggregated into a network-level score via summation (Abdelfattah et al., 2021). At a more coarse-grained perspective, while there has been an increase in research on combining zero-cost metrics (Chen et al., 2021; Krishnakumar et al., 2022; White et al., 2021), this field is still largely unexplored. In this work, we attempt to tackle both levels of composition: composing saliencies into metrics and metrics into combined proxies.

We begin, in Section 3.1, by comparing a set of existing metrics which we initially believe are ripe for combination. These metrics are divided into two groups: one focusing on the statistics of gradient-based information, and the other centering on the statistics of activation patterns. In Section 3.2, we explore what the notions of compressibility and layer-wise partitions have to offer gradient- and activation-based metrics. This is motivated by a reframing of the factors that contribute to the success of ZiCo (Li et al., 2023). By generalising these tenets, we draw links with

NASWOT and thus produce a useful and informative NASWOT variant. In Section 3.3, we motivate and provide an example of direct metric composition. This is applied to the metrics that we created in section 3.2. Finally, in Section 4, we evaluate the performance of our proposed metrics across a number of search spaces. Our main contributions are summarised as follows:

- We reframe ZiCo and enhance existing techniques for aggregating saliency scores, such as SNIP and Synflow, through the lens of network compressibility.

- We generalise the tenets of this approach to bolster NASWOT, an existing proxy for expressivity.

- We demonstrate the feasibility of direct metric composition and use it, together with the above contributions, to produce a state-of-the-art zero-cost proxy.

## 2 Related Work

**Gradient-based Saliency Scores.** Pruning-at-initialisation (PaI) considers whether it is possible to reduce the size of an initialised network whilst maintaining (or even improving) performance (Wang et al., 2022). One way to reduce the number of parameters in the network is by identifying a saliency (importance) metric on the parameters and removing those with the lowest scores. Lee et al. (2019) define a parameter as salient if its removal would significantly affect the loss. Wang et al. (2020) were more interested in the training dynamics of the network, defining a parameter's saliency around its affect on the gradient norm. From these respective approaches, they produce saliency scores, SNIP and GraSP. Tanaka et al. (2020) generalized the notion of synaptic saliency used by both SNIP and GraSP. Based on this generalisation, they proposed a saliency score, Synflow, that assessed the magnitude of a parameter's contribution to the flow of synaptic strengths.

**Zero-Cost NAS.** Finding low-cost metrics that can predict network performance accurately is one of the major challenges in NAS. Those that use no data or only a minimal amount of data are referred to as zero-cost metrics; they allow for large search spaces to be assessed in reasonable times. Many of the most successful zero-cost metrics have been inspired by the pruning-at-initialisation literature. Abdelfattah et al. (2021) turned saliency scores into zero-cost metrics by taking their sum over the parameters in a given network, transforming Synflow, SNIP, and GraSP into zero-cost metrics. These, along with other metrics such as ZiCo (Li et al., 2023) and NTK approaches (Chen et al., 2021; Shu et al., 2019), use gradient information to provide a proxy for a network's trainability. There are also many zero-cost scores that consider properties of the linear regions of the network: NASWOT (Mellor et al., 2021), ZenNAS (Lin et al., 2021), etc. These are sometimes interpreted as estimating the network's expressivity (Chen et al., 2021).

**Comparing and Combining Different Zero-Cost Metrics.** Recently, researchers have begun to investigate methods for combining different zero-cost metrics altogether. Chen et al. (2021) were among the the first to consider this. They hypothesised that metrics corresponding to trainability and expressivity should be combined. Expressivity refers to the complexity of the function that the network can represent. Trainability refers to the network's ability to effectively and faithfully direct feedback signals to its parameters. They used the condition number of the NTK as a measure of trainability and the expected number of linear regions as a measure of expressivity. Finally, Chen et al. (2021) ranked the architectures with respect to these distinct metrics and averaged the ranks, resulting in a combined proxy, TE-NAS. Although TE-NAS was a useful first step, it was an indirect means of combining metrics and didn't ensure that its metrics were tracking different network properties. To alleviate problems like this, Krishnakumar et al. (2022) investigated the relatedness of different metrics. They evaluated the conditional entropy of metrics to demonstrate which ones combine best and provide the most complementary information. We utilise similar techniques to ensure that the metrics we consider in this paper track different aspects of the network.

## 3 Method

In the following three sections, we have the overarching goal of improving methods for aggregating parameter scores into metrics and metrics into combined proxies. We further investigate several metrics that turn out to have interesting properties.

### 3.1 Comparing Existing Metrics

We begin by investigating existing metrics, to better understand their relationship, strengths and shortcomings. In particular, our focus is on the three state-of-the-art metrics: ZiCo (Li et al., 2023), Synflow (Tanaka et al., 2020) and NASWOT (Mellor et al., 2021), and the goal is to analyse them in terms of redundant and/or complementary behaviour between them. Although they might appear quite different from each other – ZiCo utilises gradient statistics of the training data, Synflow is data-independent measure of synaptic flow and NASWOT measures diversity in the activation patterns of a network – considering their common trait of achieving strong empirical performance, we wonder if there might be a single way to think about their success. In order to do that, we begin by investigating how much these metrics overlap in scoring different networks, if we take away their common trait of tracking accuracy. This is similar to what Krishnakumar et al. attempted in their work when measuring information gain from combining different metrics (Krishnakumar et al., 2022), but also focuses on a slightly different aspect. Specifically, information gain is a direct measure of how much the addition of a metric helps in predicting accuracy and, as such, was proposed as a quantitative guide to combining different metrics. Among other things, it is not commutative. We are interested in how different metrics relate to each other on a more fundamental level, before even considering their usefulness for NAS – by design this relationship should be commutative and disregard accuracy as a common factor behind their design. Therefore, rather than relying on information gain, we propose to utilise partial correlation (Rummel, 1976) to achieve this. Specifically, partial correlation treats the task-dependent accuracy as a covariate, measuring the agreement between the rankings produced by two metrics after disregarding the common covariate, hopefully extracting a more faithful description of how the metrics are related:

$$\rho_{XY \cdot Z} = \frac{\rho_{XY} - \rho_{XZ} \cdot \rho_{YZ}}{\sqrt{(1 - \rho_{XZ}^2)(1 - \rho_{YZ}^2)}} \tag{1}$$

where X, Y represent metrics; Z represents the accuracy; and each correlation is a Kendall-Tau correlation between the architecture rankings. These partial correlations between the three metrics are illustrated by Figure 1. We can observe that ZiCo correlates quite well with Synflow, which can be explained by the fact that both rely on gradient information. On the other hand, correlation between NASWOT and Synflow is significantly lower, which is aligned with the fact that NASWOT is based on activation patterns rather than gradients. Overall, our results seem to support the existing distinction of "expressivity" (activation-based NASWOT) and "trainability" (gradient-based ZiCo and Synflow) metrics discussed in previous works (Chen et al., 2021). However, the somewhat high correlation between ZiCo and NASWOT is a bit surprising and not easily explainable with the existing literature. This further motivates us to more closely investigate similarities and differences within and between gradient- and activation-based classes, which we approach in the next section through the lens of network compression.

### 3.2 Incorporating Network Compressibility and Topology into Zero-Cost NAS

In this section, we start by looking at gradient-space compressibility. This leverages the gradient to determine whether a network effectively uses the parameters and information it has available. A new interpretation of ZiCo as a measure of gradient compressibility served as the inspiration for this viewpoint. By expanding on this concept, we derive an orthogonal form of gradient compressibility, SSNR, that can be data-independent (Synflow-SNR) and yet correlates well with ZiCo. The general

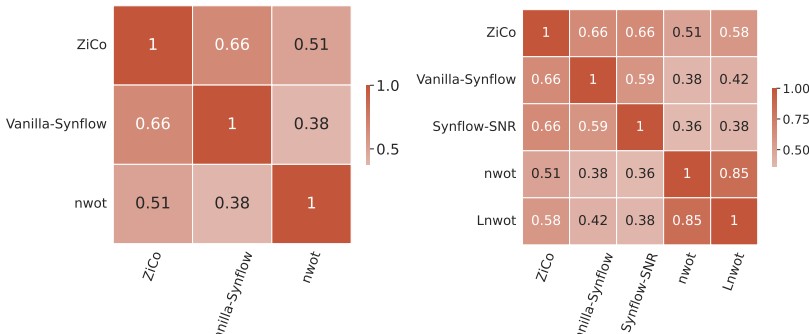

Figure 1: The partial correlation between the different metrics on NasBench201. The left image considers a small subset of existing metrics partial correlations. The right image considers a larger set of metrics, including some of those produced in Section 3.2.

tenets of compressibility and layer-wise partitions that underlie ZiCo and SSNR are used to reframe and modify NASWOT into a more informative variant.

**Gradient-space Compressibility.** Let us consider a neural network parametrised with $\Theta = \{\theta_i\}_{i=1}^M$, and a minibatch of $N$ inputs $\mathbf{X} = \{x_i\}_{i=1}^N$. The function $f : \mathbb{R}^{N \times I} \times \mathbb{R}^M \to \mathbb{R}^N$ is a composition of the network and a loss function, where the loss is applied element-wise (unreduced) to the outputs of the neural network for different elements of the input minibatch $\mathbf{X}$ and $I$ is dimensionality of a single input. Furthermore, to better relate to a network's structure, let us decompose the set of weights into its layer-wise components: $\Theta = \bigcup_{l=1}^L \Theta^l$. Training dynamics of the underlying neural network are captured by the related Jacobian matrix:

$$
\mathbf{J_X} = \begin{bmatrix} \nabla_\Theta f^1 \\ \vdots \\ \nabla_\Theta f^N \end{bmatrix} = \begin{bmatrix} \nabla_{\Theta^1} f^1 & \dots & \nabla_{\Theta^L} f^1 \\ \vdots & \ddots & \vdots \\ \nabla_{\Theta^1} f^N & \dots & \nabla_{\Theta^L} f^N \end{bmatrix} = \begin{bmatrix} \overbrace{\frac{\partial f^1}{\partial \theta_1}}^{\text{layer 1}} & \dots & \dots & \dots & \overbrace{\frac{\partial f^1}{\partial \theta_M}}^{\text{layer L}} \\ \vdots & & \ddots & & \vdots \\ \frac{\partial f^N}{\partial \theta_1} & \dots & \dots & \dots & \frac{\partial f^N}{\partial \theta_M} \end{bmatrix} \tag{2}
$$

where $f^i$ is the $i$-th output of the function (loss of the $i$-th element in $\mathbf{X}$). ZiCo fixates on how the relationship between the column-wise expectation and variance of $J_x$ relates to a network's generalizability and convergence rate. Despite achieving strong empirical results, their interpretation and link with theoretical analysis have certain shortcomings. *1)* It suggests that the metric is related to the convergence speed and that it is beneficial for predicting accuracy. However, we know there exist networks that converge slower to better results – ZiCo can correctly rank them despite the backing theory suggesting that those converging faster should be preferred. *2)* The theoretical analysis is performed only for simple networks and it is unclear how applicable it is to networks of practical size. We propose an alternative interpretation of ZiCo's strong empirical performance.

Specifically, we would argue that ZiCo effectively measures the compressibility of the matrix $\mathbf{J_X}$ along the columns. It fixes a parameter and calculates the expectation ($\mu$) and variance ($\sigma^2$) of the gradient's size over the samples. By aggregating $\mu/\sigma$ for each parameter, it assesses the compressibility of the gradient. The gradient is compressible in relation to the samples if the variance is large, as the training dynamics are likely dominated by a small subset of the inputs, specifically those that produce the largest gradient size. Similarly, a smaller variance implies that the gradient's size for each parameter is invariant over the inputs, thus ensuring the network effectively uses all the data it has available. This insight may explain the success of ZiCo as a test for whether the training dynamics use the data-set as a whole. Importantly, a compression-based interpretation – in its generalized form – is independent from either training data or performing column-wise operations, suggesting that it can be beneficial in a broader range of cases.

In order to verify our hypothesis, we consider a generalized family of zero-cost proxies, based around the concept of the compressibility of a saliency matrix. Specifically, ZiCo can be seen as taking a single instance of parameter saliency, $S(\theta_i) = \nabla_{\theta_i} f$. However, more generally, a matrix of saliency scores represents the sensitivity of the output to the parameters when evaluated on a mini-batch of data, with the main differences stemming from what loss or input is used to calculate it, or additional element-wise transformations applied to the matrix (e.g., absolute value). Irregardless of these differences, the goal of many proxies is to reduce the information contained in this matrix to a scalar value representative of a network's performance. Most existing approaches (Abdelfattah et al., 2021) simply sum the elements of this matrix, whereas we consider statistical properties related to compressibility, along the rows or the columns of our saliency matrix. As previously discussed, ZiCo probes the compressibility over the columns. Our metric, SSNR, assesses the compressibility over the rows. The formula for SSNR is as follows[1] - we breakdown its components and their corresponding reasoning below:

$$S_n^l = \sum_{\theta_i \in \Theta^l} S(\theta_i) \ \text{ and } \ \sigma^l = \sqrt{\frac{1}{|\Theta^l|} \sum_{\theta_i \in \Theta^l} \left( S(\theta_i) - \frac{S_n^l}{|\Theta^l|} \right)^2} \tag{3}$$

$$SSNR = \sum_l \frac{S_n^l}{\sigma^l} \tag{4}$$

Firstly, we will justify the use of the signal-to-noise ratio in 4. In the pruning-at-initialization literature, the saliency, $S(\theta_i)$, of a parameter, $\theta_i$, is a signal that measures how much each parameter contributes to the final model accuracy. If two architectures have similar total saliency strengths, $(S_n)$, the one whose parameters have more diverse saliency signals has a better chance of being pruned without harming accuracy, i.e., higher compressibility. During pruning, the parameters with lower saliency scores will be dropped, but the model accuracy won't necessarily be strongly affected. This means that there exist other models that have a lower overall saliency $(S_n)$, with a similar ground truth accuracy, illustrating a shortcoming of summing saliencies to score architectures. Intuitively, a high-variance saliency saliency over the parameters indicates that only a small subset of the parameters are highly important and the rest are somewhat redundant - potentially preventing the network from learning effectively.

Secondly, we must justify the layer-wise partition in 4. Each layer in the network will receive an equal amount of total synaptic saliency (Tanaka et al., 2020). This is crucial to understand because layers of varying widths will spread the saliency differently, affecting the noise. This justifies the use of a layer-wise partition as the noise (standard deviation) we wish to extract is within layers, not between them.

In summary, ZiCo measures the compressibility of the gradient across the columns, ensuring that all inputs are used effectively in the training dynamics. Similarly, SSNR measures compressibility of synaptic saliency across the rows, ensuring that all parameters are used effectively in the training dynamics. Sometimes we will distinguish the measure of saliency that SSNR acts on, and we will refer to these SSNR instances as Synflow-SNR, SNIP-SNR, and GraSP-SNR.

**Activation-space compressibility.** We have shown that the success of ZiCo and SSNR can be attributed to the fact that both approaches consider similar concepts of gradient spread (which we call compressibility) with a form of layer-wise granularity. Considering our pilot study presented in Section 3.1 which showed that ZiCo and NASWOT share similarities in their scoring behaviour, we now ask the question if a similar notion of compressibility could have an interpretation with respect to activation patterns? To answer this question, we realise that activation patterns are maximally compressed when they are maximally orthogonal, as this ensures that the linear regions

---

[1]Applying a logarithm to $S_n^l/\sigma^l$, much as ZiCo formulated, improves performance on some search spaces - albeit not on all of them. We consider this in more detail in Section 4.

are distributed over the inputs and thus are effectively utilising all available neurons. Mellor et al. (2021) use the determinant to assess whether the activation patterns are compressed over the input space (decorrelated over a mini-batch), and as such, whether all neurons provide novel information. This is motivated by the assumption that inputs with decorrelated activation patterns will be easier for the network to separate. Overall, we would argue that, by design, NASWOT is a direct measure of compressability in the activation space. However, NASWOT doesn't consider a key part of what bolstered ZiCo and SSNR: a layer-wise partition. Might such a partition be sensible also in its case?

On a practical basis, network-level NASWOT kernels are untenable if batches are reshaped or subdivided between layers (Duan et al., 2021); arguably, this alone justifies the need for layer-wise partition. Furthermore, the partition is easily interpretable, with the layer-wise NASWOT values representing the 'practical expressivity' that each layer contributes to the network. For a NASWOT-Kernel $K$, Layer-wise NASWOT (Lnwot) assigns each layer a score of $log|K^l|$, where $K^l$ represents the layer-wise NASWOT-Kernel. It then sums them into a final metric, where the log's on each layer improve computational stability.

## 3.3 Directly Combining Compressibility Metrics

In the previous two sections we have explored improvements to existing metrics. In this section, we explore means for combining a gradient-based metric (SSNR) with an activation-based approach (NASWOT), pursuing a broader objective: how to faithfully combine any two chosen metrics. These combinations may be direct, acting on network-, layer-, or even parameter- level scores, or indirect, acting on architecture rankings. As an approach to metric combination, Chen et al. (2021) averaged the ranks of the architectures with respect to the constituent metrics, an approach that we will abbreviate as the "rank average" or "RankAve". RankAve has the advantage that it equally weights the contribution of each metric, this is empirically demonstrated in 1. However, it also has a number of drawbacks. Combining ranks is feasible for some search algorithms and search spaces, however is intractable in many examples (Lin et al., 2021). Furthermore, it is indirect and arguably doesn't create a combined metric, instead offering a method to combine metrics. These insights inspire a general goal of discovering more direct and flexible means to combine metric scores. In this section, we consider a first instance of this more general goal before applying it to distinct metrics of interest, SSNR and Lnwot.

**Direct Composition.** In order to assess potential forms of a combined metric, define two arbitrary metrics, $\tau_i$ and $\tau_j$. We will assume these metrics are independent random variables, in which samples of the variable come from evaluating it on a network. We define, for $k \in \{i, j\}$, $\mu_k = \mathbb{E}[\tau_k]$ and $\sigma_k^2 = Var(\tau_k)$. What operation can we apply to compose these two metrics such that neither dominates the variance? The most natural binary operation is addition. Clearly, $Var(\tau_i + \tau_j) = \sigma_i^2 + \sigma_j^2$. But what is the effect of the variance on the rankings? Suppose that $\sigma_i \gg \sigma_j$, then (Itô, 1984):

$$P(|(\tau_i + \tau_j) - (\mu_i + \mu_j)| \geq k) \leq \frac{\sigma_i^2 + \sigma_j^2}{k^2} = \mathcal{O}(\sigma_i^2) \tag{5}$$

This indicates that the distributional properties of $\tau_i + \tau_j$ are dominated by $\tau_i$ and hence the overall ranking of the architectures is dominated by $\tau_i$. As the variances of the metrics are unlikely to be similar, the metric with a larger variance will dominate. After ruling out addition, the next simplest operation on the reals is multiplication (Goodman, 1962):

$$Var(\tau_i \cdot \tau_j) = \sigma_i^2 \sigma_j^2 + \mu_j^2 \sigma_i^2 + \mu_i^2 \sigma_j^2 = \sigma_i^2 \sigma_j^2 \left[1 + (\mu_j/\sigma_j)^2 + (\mu_i/\sigma_i)^2\right] \tag{6}$$

This indicates a more nuanced relationship between the metrics determines the rankings. Although certainly not guaranteed, if the metrics' $\mu_k$ and $\sigma_k$ scale similarly and have similar distributional characteristics, then this method prevents either metric from dominating the variance. A reasonable concern is that, even when employing metrics that exhibit the lowest correlation, assuming

independence might not be justified. In spite of the size of these assumptions, we find evidence that, for Lnwot and SSNR, the contributions of the distinct metrics to the combined scores are reasonably well-balanced (Table 1). Although there are presumably superior operations to multiplication for direct composition, this analysis of direct metric combination simply aims to serve as a proof-of-concept. We leave it to future work to further investigate methods for balancing metrics' variances in direct metric composition, potentially by considering $\left(\tau_i^\alpha \cdot \tau_j\right)$, for a suitable choice of $\alpha$.

A final point to note is that by simply multiplying the final Lnwot and SSNR scores, much of the layer-wise information that is collected by these metrics is lost. As such, we consider a layer-wise multiplication of the scores and then sum, effectively taking the dot product of the layer-wise values of Lnwot and SSNR. This layer-wise composition allows individual layers to be assessed based on their contribution to the network and incorporates the orthogonality of Lnwot and SSNR over the layers. Bringing all of this together into a single metric, we propose a *Topological*, layer-by-layer, proxy (T-CET) which balances metrics for *Compressible Expressivity* and *Trainability*:

$$\text{T-CET} = \sum_l \frac{S_n^l}{\sigma^l} \cdot log|K^l| \tag{7}$$

Table 1: In this table, A (RankAve or T-CET) denotes the method used to combine the metrics X (Synflow-SNR or SNIP-SNR) and Y (Lnwot). The first two rows show the partial correlation of the combination with X and Y respectively. RankAve is equally correlated with both metric X and Y; however, T-CET, not directly trying to do so, also roughly balances the contributions - albeit becoming slightly more correlated with X and less correlated with Y. This data is from NATSBench-TSS on CIFAR-10 (C-10), CIFAR-100 (C-100) and ImageNet-120 (IN).

| $Y \downarrow$ | $A \rightarrow$ | Rank average | | | | | | T-CET | | | | | |
| | $X \rightarrow$ | Synflow-SNR | | | SNIP-SNR | | | Synflow-SNR | | | SNIP-SNR | | |
| | $Z \rightarrow$ | C-10 | C-100 | IN | C-10 | C-100 | IN | C-10 | C-100 | IN | C-10 | C-100 | IN |
| | $\rho_{XA_{XY} \cdot Z}$ | 0.80 | 0.80 | 0.80 | 0.79 | 0.79 | 0.79 | 0.87 | 0.87 | 0.81 | 0.94 | 0.94 | 0.89 |
| Lnwot | $\rho_{YA_{XY} \cdot Z}$ | 0.80 | 0.80 | 0.80 | 0.80 | 0.80 | 0.79 | 0.62 | 0.61 | 0.57 | 0.59 | 0.59 | 0.58 |

## 4  Evaluation

In this section, we demonstrate a variety of strong experimental results. *1)* We perform an empirical analysis of the proposed metrics on multiple NAS benchmarks with rank correlation[2]. *2)* To demonstrate that our metrics are robust we tested them on a random sample of architectures from a variety of NASLib tasks. *3)* We show T-CET outperforms many existing zero-cost metrics on the ZenNAS EfficientNet search space with Aging Evolution search on the CIFAR task.

### 4.1  Micro Search Space Evaluation

In the following discussion, we compare SSNR and T-CET to existing metrics on Micro Search Spaces: NATS-Bench-TSS (NASBench201, Dong and Yang (2020); Dong et al. (2021)) and NASBench-1shot1 (Zela et al., 2020). As shown in Table 2, our first proposed class of metrics (SSNR) significantly boosted gradient-based metrics like Synflow and SNIP. As opposed to NATS-Bench-TSS , whose design was based on a fixed DARTS-like graph, NASBench-1shot1 defined three subsets from the original NASBench101 Ying et al. (2019). In these spaces, we search to determine both the operations on the node and how their edges are connected to the parent nodes, which brings an extra dynamic to the path complexity. In fact, on both NATS-Bench-TSS and NASBench-1shot1 the SNIP variants of SSNR and T-CET outperformed all of the existing proxies that we considered.

---

[2]For readability we only publish Kendall-Tau correlations for all the experiments

Moreover, our topological combination (T-CET) outperformed SSNR in the majority of cases for the micro search spaces.

Table 2: Performance of different zero-cost metrics on micro search space benchmarks, reported as Kendall-$\tau$ ranking correlation between scores and test accuracy of all networks in a benchmark. For benchmarks containing multiple seeds for each model, average accuracy was used. Unless specified otherwise, the same applies to other tables.

| Benchmark | NATSBench-TSS | | |
|---|---|---|---|
| Task | CIFAR-10 | CIFAR-100 | ImageNet16-120 |
| FLOPS | 0.58 | 0.55 | 0.52 |
| #Params | 0.58 | 0.55 | 0.52 |
| Grad-Norm | 0.46 | 0.47 | 0.43 |
| GraSP | 0.37 | 0.37 | 0.40 |
| Synflow | 0.54 | 0.57 | 0.56 |
| SNIP | 0.46 | 0.46 | 0.43 |
| ZiCo | 0.61 | 0.61 | 0.60 |
| Zen-Score | 0.29 | 0.28 | 0.29 |
| NASWOT | 0.58 | 0.62 | 0.60 |
| Lnwot | 0.57 | 0.57 | 0.56 |
| SSNR and T-CET | | | |
| Synflow-SNR | 0.63 | 0.60 | 0.59 |
| SNIP-SNR | 0.68 | **0.65** | **0.63** |
| T-CET(Synflow) | 0.65 | 0.62 | 0.58 |
| T-CET(SNIP) | **0.69** | **0.65** | 0.62 |

| Benchmark | NB-1shot1 | | |
|---|---|---|---|
| Search Space | SS1 | SS2 | SS3 |
| FLOPS | 0.56 | 0.60 | 0.51 |
| #Params | 0.57 | 0.61 | 0.52 |
| NASWOT | 0.45 | 0.50 | 0.48 |
| Lnwot | 0.52 | 0.57 | 0.50 |
| Synflow | 0.52 | 0.51 | 0.36 |
| SNIP | -0.08 | -0.05 | -0.09 |
| SSNR and T-CET | | | |
| Synflow-SNR | 0.52 | 0.60 | 0.49 |
| SNIP-SNR | 0.58 | 0.62 | 0.52 |
| T-CET(Synflow) | 0.55 | 0.62 | 0.51 |
| T-CET(SNIP) | **0.60** | **0.64** | **0.54** |

## 4.2 Macro Search Space Evaluation

The picture is more complex in the Macro-Search spaces than the Micro-Search spaces. In NATS-Bench-SSS, Synflow and SNIP perform particularly well. However, SSNR performs rather well in NATS-Bench-SSS, either significantly outperforming or slightly under-performing the best existing metrics. In these cases T-CET is hampered by its consideration of NASWOT, which helps to explain its inferior performance. On NASBench-Macro (Su et al., 2021) and BLOX (Chau et al., 2022), which are particularly challenging search spaces, SSNR and T-CET were competitive with the best pre-existing metrics.

Table 3: Performance of different zero-cost metrics on macro search space benchamarks.

| Benchmark | NATSBench-SSS | | |
|---|---|---|---|
| Task | CIFAR-10 | CIFAR-100 | ImageNet16-120 |
| FLOPS | 0.44 | 0.19 | 0.41 |
| #Params | 0.69 | 0.53 | 0.68 |
| Grad-Norm | 0.35 | 0.34 | 0.49 |
| GraSP | -0.09 | 0.01 | 0.29 |
| Synflow | 0.61 | **0.60** | 0.39 |
| SNIP | 0.42 | 0.46 | 0.57 |
| ZiCo | 0.54 | 0.55 | 0.70 |
| Zen-Score | 0.50 | 0.52 | 0.69 |
| NASWOT | 0.45 | 0.43 | 0.42 |
| Lnwot | 0.61 | 0.31 | 0.49 |
| SSNR and T-CET | | | |
| Synflow-SNR | **0.75** | 0.56 | **0.76** |
| SNIP-SNR | **0.75** | 0.56 | **0.76** |
| T-CET(Synflow) | **0.75** | 0.44 | 0.66 |
| T-CET(SNIP) | **0.75** | 0.45 | 0.66 |

| Benchmark | BLOX | NB-Macro |
|---|---|---|
| FLOPS | 0.38 | 0.56 |
| #Params | **0.40** | 0.22 |
| NASWOT | -0.00 | **0.63** |
| Lnwot | -0.00 | 0.59 |
| Synflow | -0.31 | 0.52 |
| SNIP | -0.35 | 0.50 |
| SSNR and T-CET | | |
| Synflow-SNR | 0.39 | 0.52 |
| SNIP-SNR | **0.40** | 0.52 |
| T-CET(Synflow) | **0.40** | 0.55 |
| T-CET(SNIP) | 0.34 | 0.55 |

### 4.3 Random Sampling - NDS and NASLib

To verify that our approaches are robust on different tasks and design spaces, we evaluate our methods on NASLib (Ruchte et al., 2020) by randomly sampling architectures from diverse search spaces and tasks, spanning scene classification (Cl-S), object classification (Cl-O) and Jigsaw (Jig) tasks from TransNAS-Bench-101-Micro (TB-101, Duan et al. (2021)), and CIFAR-10 from NAS-Bench-301 (NB-301, Zela et al. (2022)). Specifically, we randomly sampled 500 architectures from each task and evaluated the success of a selection of zero-cost proxies for these architectures. For evalution on NDS (Radosavovic et al., 2019), we implemented the same procedure with the only difference being that the sampling is pre-done. In 9 out of the 12 task-space pairs shown in Table 4, either SSNR or T-CET outperforms the best displayed pre-existing metric. Furthermore, each of the pre-existing metrics have at least one instance where they perform significantly weaker than the other metrics, as opposed to SSNR and T-CET which are much more stable across evaluations. Finally, particularly in the NDS search space, the topological combinations show consistent improvements over their SNR counterparts, with the only exception being their deployment on ResNet.

Table 4: Performance of different zero-cost proxies on tasks from NDS and NASLib. For NASLib (TB-101 and NB-301), a random sample of 500 networks was used to estimate ranking correlation. For NDS, we used the standard pre-determined samples.

| Benchmark | TB-101 | | | NB-301 | NDS | | | | | | | |
|---|---|---|---|---|---|---|---|---|---|---|---|---|
| | Cl-S | Cl-O | Jig | C-10 | Darts | Darts-fix | NASNet | ENAS | Amoeba | PNAS | ResNet | Resnextb-b |
| FLOPS | 0.45 | 0.32 | 0.29 | 0.37 | 0.50 | 0.02 | 0.29 | 0.41 | 0.24 | 0.39 | 0.33 | 0.34 |
| #Params | 0.44 | 0.32 | 0.29 | **0.41** | 0.49 | 0.02 | 0.29 | 0.41 | 0.24 | 0.39 | 0.59 | 0.46 |
| Synflow | 0.54 | 0.42 | 0.33 | 0.13 | 0.30 | -0.09 | 0.02 | 0.13 | -0.06 | 0.18 | 0.14 | 0.43 |
| SNIP | 0.48 | **0.49** | 0.38 | 0.03 | 0.27 | -0.13 | -0.05 | 0.09 | -0.09 | 0.15 | 0.26 | 0.45 |
| ZiCo | 0.45 | 0.32 | 0.29 | 0.37 | 0.34 | 0.09 | 0.09 | 0.20 | -0.02 | 0.19 | 0.14 | 0.47 |
| SSNR and T-CET | | | | | | | | | | | | |
| Synflow-SNR | 0.57 | 0.44 | 0.39 | 0.31 | 0.53 | 0.10 | 0.29 | 0.41 | 0.25 | 0.39 | 0.59 | 0.48 |
| SNIP-SNR | **0.58** | 0.43 | 0.40 | 0.36 | 0.54 | 0.16 | 0.29 | 0.41 | 0.24 | 0.39 | **0.60** | 0.48 |
| T-CET(Synflow) | 0.57 | 0.44 | 0.40 | 0.34 | 0.54 | 0.15 | **0.34** | 0.44 | **0.28** | 0.41 | 0.32 | **0.54** |
| T-CET(SNIP) | 0.56 | 0.47 | **0.41** | 0.32 | **0.56** | **0.20** | **0.34** | **0.45** | 0.27 | **0.42** | 0.34 | **0.54** |

### 4.4 Practical Search Space

Moving beyond the more theoretically driven search spaces we considered a practical example. To align with previous works, we used a search space consisting of residual blocks and bottleneck blocks as defined in ResNet, which is aligned with ZenNAS (Lin et al., 2021). Table 5 demonstrates that the metrics which include an estimate of gradient compressibility, T-CET and ZiCo, can even outperform a metric (Zen-Score) designed with this search space in mind.

Table 5: Top-1 accuracies for zero-cost proxies on ZenNAS Search-Space. Budget: model size N<1M. 'Random':average accuracy ± std for a random searched 32 networks.

| Method | Random | FLOPs | Grad-Norm | Synflow | NASWOT | TE-Score | Zen-Score | ZiCo | T-CET(Synflow) | T-CET(SNIP) |
|---|---|---|---|---|---|---|---|---|---|---|
| CIFAR-10 (Acc.%) | 93.5±0.7 | 93.1 | 92.8 | 95.1 | 96.0 | 96.1 | 96.2 | 97.0 | 96.6 | **97.2** |
| CIFAR-100 (Acc.%) | 71.1±3.1 | 64.7 | 65.4 | 75.9 | 77.5 | 77.2 | 80.1 | 80.2 | 80.0 | **80.4** |

### 4.5 Logarithm Ablation

Inspired by Li et al. (2023), we performed an ablation to see whether including a logarithm around the signal to noise ratio improved SSNR's performance. We observed that in some cases (NASBench-Macro, NATS-Bench SSS, and Resnext-b) the logarithm did improve proxy performance however in most cases it did not. We hypothesize that this difference in the effect of the logarithm is related to

the complexity of search spaces or distributional characteristics of the proxies that the log helps to stabilize. We leave it to future work to further investigate this phenomena and all the results given in this paper simply implemented the more suitable (higher performing) case for each search space. We publish our full ablation results in the appendix A.

# 5 Conclusion

The main contributions of our work have been to investigate methods for converting parameter-level scores into dependable and accurate zero-cost proxies. We have extended the existing proxy methods (SSNR) and analyzed the fundamental principles of these enhancements. In doing so, we have furthered understanding about the underlying principles of metrics such as ZiCo and NASWOT. Additionally, with this novel compressibility framework we were motivated to directly combine proxies for gradient-based and activation-based compressibility to produce T-CET. Lastly, we showcased the state-of-the-art empirical performance of our metrics across a wide range of NAS benchmarks.

# 6 Broader Impact Statement

After careful reflection, the authors have determined that this work presents no notable negative impacts to society or the environment.

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

Table A.1: Tables displaying the effect of the log around the signal-to-noise ratio on the performance of SSNR, for the majority of search spaces considered in the main paper.

| Benchmark | NDS | | | | | | | |
|---|---|---|---|---|---|---|---|---|
| | Darts | Darts-fix | NASNet | ENAS | Amoeba | PNAS | ResNet | Resnextb-b |
| Synflow-SNR | 0.53 | 0.10 | 0.29 | 0.41 | 0.25 | 0.39 | 0.59 | 0.45 |
| Log-Synflow-SNR | 0.35 | 0.08 | 0.09 | 0.20 | -0.01 | 0.20 | 0.15 | 0.48 |
| SNIP-SNR | 0.54 | 0.16 | 0.29 | 0.41 | 0.24 | 0.39 | 0.60 | 0.49 |
| Log-SNIP-SNR | 0.35 | 0.09 | 0.09 | 0.21 | -0.01 | 0.20 | 0.16 | 0.48 |

| Benchmark | NATSBench-SSS | | |
|---|---|---|---|
| Task | CIFAR-10 | CIFAR-100 | ImageNet16-120 |
| Synflow-SNR | 0.69 | 0.53 | 0.67 |
| Log-Synflow-SNR | 0.75 | 0.56 | 0.76 |
| SNIP-SNR | 0.69 | 0.53 | 0.68 |
| Log-SNIP-SNR | 0.75 | 0.56 | 0.76 |

| Benchmark | BLOX | NB-Macro |
|---|---|---|
| Synflow-SNR | 0.39 | 0.22 |
| Log-Synflow-SNR | -0.34 | 0.52 |
| SNIP-SNR | 0.40 | 0.23 |
| Log-SNIP-SNR | -0.23 | 0.52 |

| Benchmark | NATSBench-TSS | | |
|---|---|---|---|
| Task | CIFAR-10 | CIFAR-100 | ImageNet16-120 |
| Synflow-SNR | 0.63 | 0.60 | 0.59 |
| Log-Synflow-SNR | 0.60 | 0.60 | 0.59 |
| SNIP-SNR | 0.68 | 0.65 | 0.63 |
| Log-SNIP-SNR | 0.63 | 0.63 | 0.62 |

| Benchmark | NB-1shot1 | | |
|---|---|---|---|
| Search Space | SS1 | SS2 | SS3 |
| Synflow-SNR | 0.52 | 0.60 | 0.49 |
| Log-Synflow-SNR | 0.50 | 0.57 | 0.48 |
| SNIP-SNR | 0.58 | 0.62 | 0.52 |
| Log-SNIP-SNR | 0.52 | 0.58 | 0.50 |

Table B.1: Experiments cost in GPU hours to reproduce our results, the cost includes scoring all relevant networks (see the main paper for information on how many are used) from a search space using T-CET(SNIP). For ZenNAS search space, we strictly followed the searching and training pipeline in the original paper. It takes approximately 30 hours to search, while the remaining time is for model training.

| Experiment | NATSBench | | NB-1shot1 | | | BLOX | NB-Macro | TB-101 | NB301 | NDS | ZenNAS |
|---|---|---|---|---|---|---|---|---|---|---|---|
| | TSS | SSS | SS1 | SS2 | SS3 | | | | | | |
| Cost (GPU hours) | 4.2 | 6.4 | 2.2 | 7.3 | 41.7 | 12.2 | 3 | 4 | 4 | 72 | 134.7 |

## A  Ablation on logarithm

The results in Table A.1 demonstrate that the log around the signal-to-noise ratio generally hinders performance. However, in some cases it proves beneficial. This finding is as yet only empirical as we leave it to future work to produce a theory of the effects of re-scalings (like the log) on the performance of zero-cost proxies.

## B  Cost of running experiments

All experiments were run on a single NVIDIA GeForce RTX 3090. We present the cost of reproducing our experiments in GPU hours in the table B.1.

