# OpenReview forum: "Exploiting Network Compressibility and Topology in Zero-Cost NAS"
_automl.cc/AutoML/2023/Conference — AutoML 2023 MainTrack_

### Official Review · Reviewer_Qh4i · 2023-03-28

**Potential Impact On The Field Of Automl Rating:** 3
**Technical Quality And Correctness:** The paper is technically focused and …
**Technical Quality And Correctness Rating:** 3
**Clarity:** 1. Kindly ask the author to provide t…
**Clarity Rating:** 3
**Actions Required To Increase Overall Recommendation:** Please refer to the above section.

**Summary Of Contributions:**

The paper discusses Neural Architecture Search (NAS) to discover high-performance neural network architectures. However, current NAS approaches can be costly, requiring extensive evaluations or training of large super networks. Zero-cost proxies have been proposed to predict performance efficiently to reduce search costs, but little attention has been given to their mutual effects. The paper analyzes existing proxies through network compressibility, trainability, and expressivity and designs a new saliency and metric aggregation method informed by these factors, leading to state-of-the-art results across popular NAS benchmarks.

**Overall Review:**

The proposed method in the paper appears to be a unique approach to combining different zero-cost NAS (ZC-NAS) methods for evaluating neural architectures. However, the results achieved are only marginally better or sometimes worse than those of single ZC-NAS methods, which raises some concern. Additionally, the paper mentions TE-NAS in the related work section but does not provide further details about it in other parts of the paper, which could be confusing.

It is also worth noting that the paper "Generic Neural Architecture Search via Regression" presents a near zero-cost method with significantly better results. This suggests that while the approach in the current paper is interesting, there may be other more effective methods for zero-cost NAS that the authors could consider exploring.

In general, I believe this is a well-written paper. However, if the authors could improve the significance of their results, I would be inclined to give it a higher rating.


**Potential Impact On The Field Of Automl:**

The paper's approach to ZC-NAS, which involves revisiting and analyzing existing proxies and viewing them through a common lens of network compressibility, trainability, and expressivity, has the potential to advance the field of automated machine learning. The proposed saliency and metric aggregation method informed by compressibility, orthogonality, and network topology could improve the accuracy and efficiency of NAS, reducing search costs and time. The marginally better results achieved across various NAS benchmarks demonstrate the potential impact of this approach.

**Reproducibility (Optional):**

The authors have provided the code, and given that it is based on previous works, it is highly likely that the code is reliable. However, I have not personally verified its accuracy.

**Review Confidence:**

3: You are fairly confident in your assessment. It is possible that you did not understand some parts of the submission or that you are unfamiliar with some pieces of related work.

**Review Rating:**

7: Weak Accept: Technically sound paper with moderate-to-high impact and strong evaluation, with perhaps some minor flaws.

**Review Summary:**

Please refer to the above section.

---

> ### Author Response · Authors · 2023-05-01
> **Thank you for your support of our work**
>
> We thank the reviewer for the positive feedback, and appreciate the time and effort that the reviewer has shown to help us improve our work. We have further updated our paper and hope to have incorporated all your suggestions satisfactorily below.
>
>
> >  Kindly ask the author to provide the topology in Appendix for searched structure in the ZenNAS search space.
>
> We will add that searched architecture in the appendix.
>
> T-cet(SNIP): `SuperConvK3BNRELU(3,64,1,1)SuperResK1K3K1(64,152,1,40,5)SuperResK1K3K1(152,128,1,48,5)SuperResK1K5K1(128,256,2,24,4)SuperResK1K5K1(256,40,2,16,1)SuperResK1K5K1(40,80,2,32,1)SuperConvK1BNRELU(80,64,1,1)`
>
> T-cet(Synflow):
> `SuperConvK3BNRELU(3,176,1,1)SuperResK1K3K1(176,152,1,48,4)SuperResK1K3K1(152,152,1,32,5)SuperResK1K3K1(152,240,1,32,4)SuperResK1K3K1(240,112,2,48,1)SuperResK1K3K1(112,120,2,16,1)SuperResK1K3K1(120,112,2,40,1)SuperConvK1BNRELU(112,96,1,1)`
>
> Zico:
> `SuperConvK3BNRELU(3,96,1,1)SuperResK1K3K1(96,80,1,48,1)SuperResK1K3K1(80,192,2,48,5)SuperResK1K3K1(192,64,2,40,1)SuperResK1K3K1(64,192,1,32,5)SuperResK1K3K1(192,96,2,40,4)SuperConvK1BNRELU(96,80,1,1)`
>
> > Please consider making all the terms consistent with the original paper, such as NASWOT (instead of naswot)
>
> We thank the reviewer for highlighting this inconsistency and have amended it for the camera-ready version. We are also amending inconsistent notation for FLOPS and GraSP in the tables.
>
> > However, the results achieved are only marginally better or sometimes worse than those of single ZC-NAS methods, which raises some concern.
>
> As explained further in our response to reviewer 6bMH, our paper provides two distinct contributions. Our first proposed class of metrics is SSNR, which is a single metric that often significantly outperforms SOTA metrics. Although T-CET only produces a modest improvement upon SSNR, it significantly outperforms pre-existing SOTA and provides a proof-of-concept for direct metric combination, an as yet unexplored area of research.
>
> > Additionally, the paper mentions TE-NAS in the related work section but does not provide further details about it in other parts of the paper, which could be confusing.
>
> We respectfully disagree with this account. TE-NAS is introduced in the related work section for two reasons. Firstly, it attempts to combine metrics for expressivity and trainability. Secondly, it is an example of an indirect means of metric combination from which our method is proposed in response to. With regards to the first point, towards the end of Section 3.1 we reconsider ‘trainability’ and ‘expressivity’ directly citing TE-NAS. With regards to the second point, when introducing the indirect combination method RankAve in Section 3.3 we directly cite TE-NAS.
>
> > It is also worth noting that the paper "Generic Neural Architecture Search via Regression" presents a near zero-cost method with significantly better results. This suggests that while the approach in the current paper is interesting, there may be other more effective methods for zero-cost NAS that the authors could consider exploring.
>
> We credit GenNAS as being an insightful and important contribution to the field of zero-cost NAS, however its focus is somewhat different to the motivations behind our paper, and zero-cost metrics more generally. GenNAS predicts a network's ability, on a range of tasks, by considering the ability of extracted tensors from its intermediate layers to regress towards signal bases. It provides a strong performance predictor of the network’s ability to perform a variety of difficult tasks via considering its performance on a generic low-cost proxy task. This however does not provide insight into why a given network performs well, beyond its ability to perform well on the generic tasks considered by GenNAS.
>
> In contrast, zero-cost proxies extract more fundamental properties of the network, be that synaptic flow, gradient variance, etc. These likely can’t compete on performance with methods like GenNAS, but this sacrifice allows for interpretability. Zero-cost metrics implicitly provide an explanation for why a given network is performing well. The narrative of this paper is not that SSNR and T-CET outperform all (near) zero-cost methods but that they outperform other common zero-cost metrics and provide an interpretable overarching explanation for network success in terms of (layer-wise) compressibility be that gradient- or activation- based.

---

### Official Review · Reviewer_2i84 · 2023-04-08

**Potential Impact On The Field Of Automl Rating:** 4
**Technical Quality And Correctness Rating:** 3
**Clarity:** The paper is well-written and well-or…
**Clarity Rating:** 4

**Summary Of Contributions:**

The article discusses Neural Architecture Search (NAS) and how it has been successful in discovering high-performance neural network architectures. However, current NAS approaches require extensive evaluation or training, so zero-cost proxies have been proposed to efficiently predict architecture performance. The article argues that existing proxies should be revisited and analyzed through the lens of network compressibility, trainability, and expressivity. By doing so, a better understanding of the high-level relationship between different proxies can be built and some can be refined. The article proposes a novel saliency and metric aggregation method informed by compressibility, orthogonality, and network topology that yields state-of-the-art results in popular NAS benchmarks.

**Actions Required To Increase Overall Recommendation:**

I am open to further discussions, and willing to adjust my score if the author could respond my questions above.

**Overall Review:**

In general, this paper will be interesting to people working in NAS and AutoML.
I have voiced my concerns above.

**Potential Impact On The Field Of Automl:**

This work will be of high interest to the AutoML community. It studies recently proposed training-free indicators for NAS, and proposed a new proxy with reasonable explanations.

**Review Confidence:**

4: You are confident in your assessment, but not absolutely certain. It is unlikely, but not impossible, that you did not understand some parts of the submission or that you are unfamiliar with some pieces of related work.

**Review Rating:**

7: Weak Accept: Technically sound paper with moderate-to-high impact and strong evaluation, with perhaps some minor flaws.

**Review Summary:**

This paper provides a concrete analysis of the new proxy it proposes (which is very interesting), and conducts comprehensive experiments.
I vote weak accept. I hope the author could address my concerns above.

**Technical Quality And Correctness:**

I have the following concerns:

1. Where is $K^l$ defined (line 223)? Maybe I missed something?
2. In addition to the explanation on page 7, are there any ablation studies on different ways to combine S and K in Eq. 8? E.g. with or without logarithm?
3. The novelty is slightly limited, as both Gradient (SynFlow over variance) and Action-space (layer-wise Naswot) Compressibility are proposed by existing works or modified versions.

---

> ### Author Response · Authors · 2023-05-01
> **Thank you for your support of our work**
>
> We thank the reviewer for the positive feedback, and appreciate the time and effort that the reviewer has shown to help us improve our work. We have further updated our paper and hope to have incorporated all your suggestions satisfactorily below.
>
> > … Where is K^l defined(line223)
>
> This definition was indeed lacking and we have clarified this for the camera-ready version.
>
> > In addition to the explanation on page 7, are there any ablation studies on different ways to combine S and K in Eq. 8? E.g. with or without logarithm.
>
> We are indebted to this recommendation. In T-CET there are two log’s: one on the determinant (NASWOT) and one on the signal to noise ratio (SSNR). The logdet on the NASWOT kernel is required to prevent data-overflow, thus is a non-negotiable. The log on the signal to noise ratio was motivated by 2 considerations: to make SSNR’s formulation consistent with ZiCo and to stabilise the computation such that no layer dominates the score.
>
> Upon performing an ablation study we find that the logarithm does improve performance on some search spaces: NasBenchMacro, NDS-Resnext-b, NATSBench-SSS, ZenNAS - albeit not on others. We explain the disparity in the effect of the logarithm on search space results by the variance in layer-size. We believe that search spaces containing networks exhibiting significant variation in layer width, ‘unstable search spaces’, require the log to stabilise the results. By its very nature, the logarithm effectively replaces layer-wise addition of scores with layer-wise multiplication. This prevents the layers with the largest values from completely dominating the final score, even if some layers’ values are exponentially larger than others. Conversely, in search spaces with more consistent layer-widths, ‘stable search spaces’, we contend the logarithm is unnecessary. Rather than enhancing performance, the logarithm further contracts the already small differences between layer scores, making the system more susceptible to noise and consequently degrading its performance. In summary, the inclusion or exclusion of the log does not change the fundamental tenets of SSNR but merely helps to stabilise performance in some search spaces.
>
> Although these arguments for explaining the observed behaviour are not particularly rigorous, and more theory would be needed to more confidently justify our changes we can at the very least provide an empirically-justified recommendation for when to use the logarithm with SSNR. Accordingly, in the camera-ready version our table results will only stabilise SSNR with the log when the stability of the search spaces requires it.
>
> In more detail, our ablation study assessed the average Kendall tau correlation of log-less SSNR (the signal to noise ratio with no logarithm acting on it) and SSNR (as proposed in our paper) over a number of search spaces and tasks. We ranked log-less Synflow-SNR, Syflow-SNR, log-less SNIP-SNR and SNIP-SNR on a number of search spaces and datasets. On the less stable search spaces (NasBenchMacro, NDS-Resnet-b, NATSBench-SSS) the logarithm proved useful, outperforming its log-less counterpart.
>
> |                      | Resnext-b | Macro - SS1 | SSS (C10) | SSS (C100) | SSS (IN120) | Rank Average |
> |----------------------|-----------|-------------|-----------|--------------|-------------|--------------|
> | Log-less Synflow-SNR |      0.45 |        0.22 |      0.69 |       0.53 |        0.67 |          3.8 |
> | Synflow-SNR          |      0.48 |        0.52 |      0.75 |       0.56 |        0.76 |          1.8 |
> | Log-less SNIP-SNR    |      0.49 |        0.23 |      0.69 |       0.53 |        0.68 |          2.8 |
> | SNIP-SNR             |      0.48 |        0.52 |      0.75 |       0.56 |        0.76 |          1.6 |

---

> > ### Author Response · Authors · 2023-05-01
> > **Further Comments**
> >
> >
> > On the more stable search spaces (NATSBench-TSS, NASBench1shot1, BLOX and the rest of NDS) the Kendall tau correlations were higher for log-less SNR as opposed to SNR.
> >
> > |                      | Darts | Darts-fix-w-d | NASNet | ENAS | Amoeba | PNAS | ResNet | BLOX  | 1shot1 (SS1) | 1shot1 (SS2) | 1shot1 (SS3) | TSS (C10) | TSS (C100) | TSS (IN120) | RankAverage |
> > |----------------------|-------|---------------|--------|------|-------|------|--------|-------|--------------|--------------|--------------|------|------------|-------------|-------------|
> > | Log-less Synflow-SNR |  0.53 |          0.10 |   0.29 | 0.41 |   0.25 | 0.39 |   0.59 |  0.39 |         0.52 |         0.60 |         0.49 |      0.63 |       0.60 |        0.59 |        2.29 |
> > | Synflow-SNR          |  0.35 |          0.08 |   0.09 | 0.20 |  -0.01 | 0.20 |   0.15 | -0.34 |         0.50 |         0.57 |         0.48 |      0.60 |       0.60 |        0.59 |        3.86 |
> > | Log-less SNIP-SNR    |  0.54 |          0.16 |   0.29 | 0.41 |   0.24 | 0.39 |   0.60 |  0.40 |         0.58 |         0.62 |         0.52 |      0.68 |       0.65 |        0.63 |        1.14 |
> > | SNIP-SNR             |  0.35 |          0.09 |   0.09 | 0.21 |  -0.01 | 0.20 |   0.16 | -0.23 |         0.52 |         0.58 |         0.50 |      0.63 |       0.63 |        0.62 |        2.71 |
> >
> > We found similar, albeit slightly more modest, improvements for the log vs log-less variants of T-CET. We will update our tables in our paper with improved results for the camera-ready version.
> >
> > >  The novelty is slightly limited, as both Gradient (SynFlow over variance) and Action-space (layer-wise Naswot) Compressibility are proposed by existing works or modified versions.
> >
> > We disagree with this characterization of the literature. We do agree that NASWOT is, at least implicitly, described as a measure of compressibility; however, this is not the case for gradient-based compressibility. Arguments around compressibility are not used to explain the effectiveness of ZiCo and we need to be aware of any previous literature that considers the row-wise compressibility of the gradient i.e., saliency over variance. Infact, a key part of the novelty of this work derives from the fact that it generalises the tenets of some of the highest-performance metrics and from these insights generates a novel class of row-compressed metrics, SSNR.

---

### Official Review · Reviewer_6bMH · 2023-04-11

**Potential Impact On The Field Of Automl Rating:** 1
**Technical Quality And Correctness Rating:** 3
**Clarity:** The paper is well motivated and clear.
**Clarity Rating:** 4
**Actions Required To Increase Overall Recommendation:** 1. More 'random search' results to de…

**Summary Of Contributions:**

In this paper, the authors identify some existing metrics and classify them into gradient property based and activation pattern based. They identify the factors that contribute to the success of ZiCo and reformulate NASWOT to perform better. They also produce a state of the art zero cost proxy.

**Overall Review:**

The paper is well written, well motivated and easy to understand. However, the resulting scoring function (T-CET) does not deliver sufficiently improved KT/SPR to have a more profound impact on the community. Focusing on identifying more zero cost proxies and classifying them as done in Figure 1 and drawing insights to develop better zero cost proxies may help solidify the contributions of the paper, as well as more results (NDS search space can be easily initialized from their open source code for generating KT/SPR, as well as more search results demonstrating superior Top1-accuracy) would help.

**Potential Impact On The Field Of Automl:**

The paper is interesting, but the improvement in rank correlation in most cases is ~0.03, which brings into question the effectiveness of T-CET and consequently its impact in practice.

**Review Confidence:**

5: You are absolutely certain about your assessment. You are very familiar with the related work and checked all the details carefully.

**Review Rating:**

5: Borderline Leaning Reject: Technically sound paper where reasons to reject nonetheless outweigh reasons to accept. Please use sparingly.

**Review Summary:**

The paper is well written, and takes an interesting approach towards improving existing zero cost proxies. Unfortunately, the lack of a significant improvement over existing zero cost proxies, as well as lack of random search based results on more architecture spaces limits the impact of this work in practice.

**Technical Quality And Correctness:**

The papers comparision of existing metrics (from Figure 1) is interesting, and does indeed support their distinction of expressivity and trainability. However, contextualizing this comparision with more existing metrics would make the point stronger of which proxies fall in which of the introduced classification. (A larger graph with several of the existing proxies would help: synflow, jacob_cov, grad_norm, SNIP, Angle-NAS, TE-NAS, EZNAS, NASWOT, ZenNAS, ZiCo). Since the paper studies compositionality of zero cost proxies, being more thorough with such an evaluation would be valuable.

The proposed approach of developing T-CET is sound, but its resulting effectiveness is not impressive. To further support the effectiveness of their proxy, I would highly recommend testing on the NDS-CIFAR-10 search space, which is especially difficult for synflow as shown in NASWOT. It is not necessary that a higher SPR/KT would equate to better performance in search algorithms, thus, evaluating top-1 accuracies on a few more search spaces aside from the one in Table 4 may highlight the usefulness of T-CET.

---

> ### Author Response · Authors · 2023-05-01
> **Thank you for your feedback.**
>
> We thank the reviewer for the valuable and constructive comments on our work. We hope to have answered all of your questions satisfactorily below. Please do let us know if you see any further issues in the paper that are unclear or need to be addressed.
>
> > The paper is interesting, but the improvement in rank correlation in most cases is ~0.03, which brings into question the effectiveness of T-CET and consequently its impact in practice.
>
> Our paper proposes two novel types of metric: SSNR and T-CET. Firstly, we consider an alternative description for the success of ZiCo, which not only provides an interpretation of ZiCo's success that is more consistent with empirical results but also highlights a method for improving existing metrics, SSNR (which we have modified in accordance with the comments of reviewer 2i84). We find a 0.21 average Kendall Tau correlation improvement of SNIP-SNR over SNIP on NATSBench-SSS and a 0.20 average improvement on NATSBench-TSS. This demonstrates the effectiveness of the first class of metric that we propose. However, the reviewer's specific concern was the effectiveness of T-CET.
>
> This metric was motivated by our finding that gradient- and activation- based compression metrics were complementary in Section 3.1. In creating T-CET our aims were to produce a high-performance metric, robust to the gradient- or activation- based biases of various search spaces. We acknowledge that the majority of T-CET’s performance can be attributed to SSNR, however, in most of the search spaces (and datasets) that we consider T-CET outperforms SSNR, albeit if the improvement is often rather small. Moreover, it is worth highlighting to the reviewer that the aforementioned modification to SSNR (and thus T-CET) further improved T-CETs performance over existing metrics (discussed further in our response to reviewer 2i84) - later in our response we provide results showing that T-CET(SNIP) outperforms ZiCo in kendall tau correlation on average across NDS by 0.20.
>
> Furthermore, T-CET stems from a comprehensive framework encompassing a novel viewpoint on diverse high-performance metrics and a new (direct) approach to metric combination. We firmly believe that both aspects of this framework will offer valuable insights to the wider research community. A direct approach to metric combination is helpful for some search algorithms and may better discern the highest performing architectures, evidenced by our improvement over SOTA results on the top-1 ZenNAS search space. For the reasons outlined above, although we appreciate that the improvements of T-CET upon our other proposed metric class (SSNR) are modest, we believe that together SSNR and T-CET provide a significant contribution - and are the only metrics we could find, along with ZiCo, that outperform params on average over NATSBench- TSS and SSS.
>
> > The papers comparision of existing metrics (from Figure 1) is interesting, and does indeed support their distinction of expressivity and trainability. However, contextualizing this comparision with more existing metrics would make the point stronger of which proxies fall in which of the introduced classification. (A larger graph with several of the existing proxies would help: synflow, jacob_cov, grad_norm, SNIP, Angle-NAS, TE-NAS, EZNAS, NASWOT, ZenNAS, ZiCo). Since the paper studies compositionality of zero cost proxies, being more thorough with such an evaluation would be valuable.
>
> We agree that a more formal investigation into the metrics most ripe for combination is an important task. Such research is ongoing and has arguably been prompted by the work of Krishnakumar et al. referred to in our paper. This, however, was not the main focus of this paper. Section 3.1 acted to motivate our interests of understanding, generalising and combining gradient- and activation- based compressibility metrics, thus we restrict our use of metric comparison techniques to specific metric classes relevant to these aims in order to prevent overwhelming the reader - but we do direct them to similar work conducted by Krishnakumar et al. for a more comprehensive analysis.

---

> > ### Author Response · Authors · 2023-05-01
> > **Further comments**
> >
> > > The proposed approach of developing T-CET is sound, but its resulting effectiveness is not impressive. To further support the effectiveness of their proxy, I would highly recommend testing on the NDS-CIFAR-10 search space, which is especially difficult for synflow as shown in NASWOT. It is not necessary that a higher SPR/KT would equate to better performance in search algorithms, thus, evaluating top-1 accuracies on a few more search spaces aside from the one in Table 4 may highlight the usefulness of T-CET.
> >
> > We agree that the top-1 accuracies demonstrate some of the more significant contributions of this paper. As the reviewer requested, we further performed searching on ZenNAS search space with CIFAR-100 datasets and also compared our approach with more recent SOTA Zico in both CIFAR-10 and CIFAR-100 Setting:
> >
> >
> >
> > |     Method           | CIFAR-10 |CIFAR-100|
> > |----------------|------------------------------| -----------|
> > | Random         | 93.5+-0.7                    | 71.1+-3.1                     |
> > | FLOPs          | 93.1                         | 64.7                          |
> > | Grad-norm      | 92.8                         | 65.4                          |
> > | Synflow        | 95.1                         | 75.9                          |
> > | NASWOT         | 96.0                         | 77.5                          |
> > | TE-Score       | 96.1                         | 77.2                          |
> > | Zen-Score      | 96.2                         | 80.1                          |
> > | Zico           | 97.0                         | 80.2                          |
> > | T-CET(Synflow) | 96.6                         | 80.0                          |
> > | T-CET(SNIP)    | **97.2**                  | **80.43**              |
> >
> > T-CET(SNIP) is consistently finding better models than recent SOTA metrics Zen-Score and Zico in the same search settings. We hope this should help the reviewer to acknowledge the advantages of our approaches.
> >
> > Under your recommendation, we have expanded our analysis of different search spaces in NDS datasets, the results (Kendall tau correlations) are listed in the following:
> >
> > |                | Darts | Darts-fix-w-d | NASNet | ENAS | Amoeba | PNAS | ResNet | Resnext-b |
> > |----------------|-------|---------------|--------|------|--------|------|--------|-----------|
> > | Synflow        | 0.30  | -0.09         | 0.02   | 0.13 | -0.06  | 0.18 | 0.14   | 0.43      |
> > | SNIP           | 0.27  | -0.13         | -0.05  | 0.09 | -0.09  | 0.15 | 0.26   | 0.45      |
> > | ZiCo           | 0.34  | 0.09          | 0.09   | 0.20 | -0.02  | 0.19 | 0.14   | 0.47      |
> > | Synflow-SNR    | 0.53  | 0.10          | 0.29   | 0.41 | 0.25   | 0.39 | 0.59   | 0.48      |
> > | SNIP-SNR       | 0.54  | 0.16          | 0.29   | 0.41 | 0.24   | 0.39 | 0.60   | 0.48      |
> > | T-CET(Synflow) | 0.54  | 0.15          | 0.34   | 0.44 | 0.28   | 0.41 | 0.32   | 0.54      |
> > | T-CET(SNIP)    | 0.56  | 0.20          | 0.34   | 0.45 | 0.27   | 0.42 | 0.34   | 0.54      |
> >
> > We observe that our approaches significantly improved correlation in different datasets, where vanilla Synflow and SNIP correlated poorly with ground truth accuracy, which further highlights our contribution that compressibility is important for estimating model performance in zero-cost NAS.

---

> > > ### Comment · Reviewer_6bMH · 2023-05-02
> > > **Response To Authors**
> > >
> > > Thank you for your response.
> > >
> > > I appreciate that the authors have expanded their analysis of different search spaces in NDS data-set, but still believe that some key related works in the above table are missing. For instance, EZNAS significantly outperforms T-CET on several NDS design spaces. Please note that in some of the NDS search spaces, FLOPs and Params seem to be better or similar metrics for accuracy (DARTS, ENAS).
> > >
> > > I still feel that the paper is well written and well motivated and agree with the insights the paper can provide to the community, but I believe more empirical evidence is required for the results to have a more notable impact in the NAS community. Specifically, it may be very helpful to have more NAS search results to further strengthen the paper given that the KT/SPR correlation results are a little weak. The NAS search results can help us understand how these zero cost proxies are able to differentiate between the top-x% networks in a given design space, a problem highlighted in [1].
> > >
> > > Specifically, ZenNAS CIFAR-10 and CIFAR-100 correlation may be strong enough that the new experiment reported above does not lend us any insight (please correct me if I am wrong about the architecture correlation across data-sets, generally this has been true for several design spaces.)
> > >
> > > [1] Abdelfattah, M. S., Mehrotra, A., Dudziak, Ł., & Lane, N. D. (2021). Zero-Cost Proxies for Lightweight NAS.

---

> > > > ### Author Response · Authors · 2023-05-03
> > > > **Further response**
> > > >
> > > > We thank the reviewer for pointing to EZNAS as a reference paper, which significantly contributes to the field of zero-cost NAS. The main contribution of EZNAS is the generation of interpretable and generalisable zero-cost metrics by evolving more fundamental pre-existing operations with expression trees based on the statistics from existing benchmark search spaces. However, we argue that EZNAS is orthogonal to our work, which means the success of EZNAS does not conflict with our contributions, and we will list our reasons below.
> > > >
> > > > Firstly, in our work, we focus on exploring different types of compressibility and propose a unified framework to combine different compressibilities that significantly improve existing zero-cost metrics. Our methods provide unique operation types that EZNAS did not consider: a row-wise signal-to-noise ratio (SSNR). More specifically, EZNAS is limited to creating expression trees; it does not contain a stand-alone standard deviation operation and does not possess an operation for restricting a pre-existing operation to an axis of a tensor, i.e. row/column-wise. For these reasons, we believe there is, at most, a negligible chance of EZNAS finding ZiCo, SSNR, or T-CET.
> > > > In our further analysis, we find that T-CET outperforms EZNAS on all tasks in NATSBench-TSS/SSS (by up to 0.11 Spearman - shown in the table below), further suggesting the significance of the operations we propose that are unaccounted for by EZNAS.
> > > >
> > > > Moreover, we don’t believe the comparison on NDS is fair, as the EZNAS form is generated based on a fitness objective derived from NDS. Furthermore, their use of approximately 24 hours of CPU time to search for their method, as opposed to an analytic framework, arguably gives them an unfair advantage over our method. We maintain that the NAS community can bolster existing methods by considering the notion of compressibility presented in our paper; therefore, we believe that our work is orthogonal and not in competition with methods like EZNAS. We will further include those analyses in our paper.
> > > >
> > > > | Search Space (Dataset-Correlation) | TSS (C10-KT) | TSS (C100-KT) | TSS (IN120-KT) | SSS (C10-SR) | SSS (C100-SR) | SSS (IN120-SR) | DARTS (C10-KT) | Amoeba (C10-KT) | ENAS (C10-KT) | PNAS (C10-KT) | NASNet (C10-KT) |
> > > > |-------------------------------------|---------------|------------------|----------------|---------------|---------------|------------------|----------------|---------------|---------------|------------------|----------------|
> > > > | FLOPs                          	| 0.56     	| 0.54      	| 0.50       	| 0.61     	| 0.28      	| 0.58       	| 0.51       	| 0.26        	| 0.47      	| 0.34      	| 0.20        	|
> > > > | #Params                        	| 0.56     	| 0.54      	| 0.50       	| 0.87     	| 0.72      	| 0.68       	| 0.50       	| 0.26        	| 0.47      	| 0.32      	| 0.21        	|
> > > > | NASWOT                         	| 0.57     	| 0.61      	| 0.55       	| 0.45     	| 0.18      	| 0.41       	| 0.47       	| 0.22        	| 0.37      	| 0.38      	| 0.30        	|
> > > > | EZNAS                          	| 0.65     	| **0.65**  	| 0.61       	| 0.89     	| 0.74      	| 0.81       	| **0.56**   	| **0.45**    	| **0.52**  	| **0.51**  	| **0.44**    	|
> > > > | SNIP-SNR                       	| 0.68     	| **0.65**  	| **0.63**   	| **0.92** 	| **0.76**  	| **0.92**   	| 0.54       	| 0.24        	| 0.41      	| 0.39      	| 0.29        	|
> > > > | T-CET(SNIP)                    	| **0.69** 	| **0.65**  	| **0.62**   	| **0.92** 	| 0.63      	| 0.85       	| **0.56**   	| 0.27        	| 0.45      	| 0.42      	| 0.34        	|

---

> > > > > ### Author Response · Authors · 2023-05-03
> > > > > **Further response**
> > > > >
> > > > > With respect to FLOPs and #Params, although T-CET and SSNR didn’t significantly outperform these baseline metrics’ Kendall tau correlations on Amoeba, DARTS and ENAS, they provide significant gains in NASNet, PNAS and all of NATSBench. As requested, we have further considered the accuracies of the top-x models found by each metric. When considering the top-1 model accuracy on NDS, we found that one of either SNIP-SNR or T-CET(SNIP) at least matched both FLOPs and #Params on all search spaces and strictly outperformed them in five out of eight occasions. For the top-5%, T-CET(SNIP) strictly outperformed FLOPs and #Params in every space, which should lead to NAS algorithms with a more stable performance.
> > > > >
> > > > > Moreover, we found further evidence that a compressibility perspective bolsters existing metrics: in all NDS spaces that we considered, SNIP-SNR strictly outperformed SNIP at top-5% accuracy (and in all but one space for top-1 accuracy). NASWOT proved to be the strongest pre-existing metric we considered in the top-x search; however, we still strictly outperformed it on all but one occasion for top-5% accuracy and were competitive with it for top-1 accuracy.
> > > > >
> > > > > |   Search Space \ Top-1 Acc              | FLOPs     | #Params   | NASWOT    | Synflow   | SNIP  | Synflow-SNR | T-CET(Synflow) | SNIP-SNR  | T-CET(SNIP) |
> > > > > |-----------------|-----------|-----------|-----------|-----------|-------|-------------|----------------|-----------|-------------|
> > > > > | Darts      | 92.67     | 92.67     | 93.53     | **93.71** | 87.48 | 92.67       | 92.67          | 92.67     | 93.53       |
> > > > > | Darts-fix  | 92.17     | 92.17     | **92.90** | 84.99     | 2.31  | 92.17       | **92.90**      | 92.17     | **92.90**   |
> > > > > | NASNet     | 92.63     | 92.63     | **94.25** | 92.93     | 86.97 | 92.63       | 92.63          | 92.63     | 92.63       |
> > > > > | Resnext-b  | 94.19     | 94.05     | 94.05     | 94.09     | 94.18 | 94.09       | **94.48**      | 94.09     | **94.48**   |
> > > > > | ENAS       | 93.02     | 93.02     | 93.02     | **93.35** | 83.31 | 93.02       | 93.02          | 93.02     | 93.02       |
> > > > > | Amoeba     | **93.64** | **93.64** | 93.30     | 91.22     | 86.04 | **93.64**   | **93.64**      | **93.64** | **93.64**   |
> > > > > | PNAS       | 93.38     | 93.38     | **93.94** | 92.86     | 92.86 | 93.83       | 93.69          | 93.83     | 93.83       |
> > > > > | ResNet     | 93.59     | 93.59     | 92.92     | 93.26     | 92.85 | 93.59       | 92.99          | **93.70** | 92.99       |
> > > > > | #Best-Acc       | 1         | 1         | 3         | 2         | 0     | 1           | 3              | 2         | 3           |
> > > > >
> > > > > |  Search Space \ Top-5% Acc              | FLOPs     | #Params   | NASWOT    | Synflow   | SNIP  | Synflow-SNR | T-CET(Synflow) | SNIP-SNR  | T-CET(SNIP) |
> > > > > |-----------------|-----------|-----------|-----------|-----------|-------|-------------|----------------|-----------|-------------|
> > > > > | Darts     | 93.44     | 93.44     | 93.64     | 92.52     | 92.07 | 93.75       | 93.82          | 93.84     | **93.90**   |
> > > > > | Darts-fix | 91.89     | 91.91     | 92.00     | 90.87     | 89.17 | 92.09       | 92.13          | **92.23** | 92.18       |
> > > > > | NASNet    | 92.67     | 92.64     | 92.63     | 90.38     | 88.26 | 92.66       | 92.80          | 92.67     | **92.82**   |
> > > > > | Resnext-b | 93.26     | 93.58     | 93.49     | 93.52     | 93.54 | 93.59       | **93.88**      | 93.59     | **93.88**   |
> > > > > | ENAS      | 93.14     | 93.14     | 92.92     | 90.94     | 90.55 | 92.97       | 93.13          | 93.19     | **93.25**   |
> > > > > | Amoeba    | 91.24     | 91.26     | 91.14     | 88.63     | 87.32 | 91.75       | **91.95**      | 91.62     | 91.80       |
> > > > > | PNAS      | 93.50     | 93.51     | **93.83** | 93.20     | 92.85 | 93.56       | 93.59          | 93.51     | 93.58       |
> > > > > | ResNet    | 93.19     | 93.82     | 93.13     | 93.31     | 93.24 | 93.92       | **94.01**      | 93.83     | 94.00       |
> > > > > | #Best-Acc       | 0         | 0         | 1         | 0         | 0     | 0           | 3              | 1         | 4           |
> > > > >
> > > > > Our analysis highlights the advantages of SSNR and T-CET, illustrating that compressibility provides an important perspective that is largely overlooked by EZNAS’ operation vocabulary and hints at ways in which methods like EZNAS may be further improved.

---

> > > > > > ### Comment · Reviewer_6bMH · 2023-05-04
> > > > > > **Response to authors**
> > > > > >
> > > > > > Thank you for your thorough response.
> > > > > >
> > > > > > Thanks for remarking that a comparision on NDS specifically may be unfair, I agree. I also comprehend that the paper presents the idea of compressibility, which appears to strengthen existing zero-cost proxies. This could be beneficial for the community, but I'm still concerned about the extent of its impact.
> > > > > >
> > > > > > Thank you for sharing the mentioned results; over how many trials were they conducted? Generally, there can be considerable variability in NAS search results.
> > > > > >
> > > > > > I have revised my rating and will also review other discussions once they have finished to determine if I need to adjust my score further.

---

> > > > > > > ### Author Response · Authors · 2023-05-06
> > > > > > > **Thank you for your comments**
> > > > > > >
> > > > > > > Thank you for your comments. We scored all networks in the search space and reported the accuracy of the top-1 and the average of the top-5% models. Thus we only run experiments with the same initialization seeds one time. We understand that zero-cost scores can vary slightly because of random initialization and random batches, but the overall correlation should be relatively stable across different seeds. We thank for the reviewer raising this concern on the robustness of randomness; we will add an experiment for different random seeds and report accuracy in a mean +- std manner.

---

### Review · Reproducibility_Reviewer_orA8 · 2023-04-12

**Completeness Of Code And Dataset Supplement Rating:** 3
**Usability And Ease Of Reproducibility Rating:** 1

**Actions Required To Increase The Reproducibility And Overall Recommendation:**

The recommendation to the authors is to include detailed installation instructions and a requirements.txt. Furthermore, additional resources such as foresight should be included if the license allows for that and otherwise this should be noted. The bug encountered and any additional bugs should be resolved.

**Completeness Of Code And Dataset Supplement:**

A requirements.txt file is missing, so it is not clear what packages are used and how to install them.
A module named foresight is used, but it is unclear how to get it. After some google searching, it seems that  the folder foresight in https://github.com/SamsungLabs/zero-cost-nas may be the right code.
h5py needs to be installed through pip.
After installing Python and PyTorch, downloading the required datasets and performing the above, performing any of the scripts results in a long stream of the following error: find_measures() got an unexpected keyword argument 'aggregate'
Finally, README.md states that scripts/TCET_synflow_NAS_IM_flops450M.sh is for CIFAR, however it seems to be for ImageNet instead.

**Overall Reproducibility Review:**

Positive and negative aspects cannot be given at this moment, as reproducibility is mostly impossible.

**Review Confidence:**

3: You are fairly confident in your assessment. It is possible that you did not understand some parts of the submission or that you are unfamiliar with some pieces of the code or data.

**Review Rating:**

2: Strong reject, the paper appears to be completely unreproducible.

**Review Summary:**

Important elements for reproducibility are missing, see previous answers.

**Summary Of Necessary Code And Dataset Supplement:**

The paper presents a method for improving zero-cost Neural Architecture Search (NAS) by exploiting network compressibility and topology.

The authors use several existing metrics such as ZiCo (Li et al., 2023), Synflow (Tanaka et al., 2020), and Naswot (Mellor et al., 2021) to analyze their relationship, strengths, and shortcomings. They also propose new metrics such as SSNR (signal-to-noise ratio of saliency) and Lnwot (layer-wise Naswot). The authors investigate means for combining a gradient-based metric (i.e., SSNR) with an activation-based approach (Naswot) through direct composition. A novel saliency and metric aggregation method informed by compressibility, orthogonality, and network topology is introduced named T-CET, combining SSNR and Lnwot.

The authors evaluate the metrics on various image-classification benchmarks: NATS-Bench-TSS(NASBench201), NATS-Bench-SSS, NASBench-1shot1, and NASBench-Macro. They also evaluate on NASlib by randomly sampling architectures from diverse search spaces and tasks. The authors compare T-CET with other zero-cost proxies on various benchmarks with rank correlation. The datasets used in the evaluation include CIFAR-10, CIFAR-100, and ImageNet16-120.

**Usability And Ease Of Reproducibility:**

Nothing could be reproduced.
Refer to Completeness Of Code And Dataset Supplement

---

> ### Author Response · Authors · 2023-05-01
> **Thank you for you feedback**
>
> We thank you for the reviewer raising the problems in our source code; in the source code we provided, we mainly aimed to provide the requirements to reproduce the results in ZenNAS search space and provide a clear implementation for how we calculate our zero-cost metrics. We will update our code to fix the concerns that arise from the reviewer. We are sorry again that we did not provide the necessary modified version of zero-cost-nas, which is the main issue of the running failure; we have updated the repository and updated requirements.txt which exports from our environments. As the OpenReview at this stage has not accepted any revision, we will update the source code in camera-ready. We will also provide two anonymous Github links for updated ZenNAS search reproduction and a toolkit for computing our T-CET score.
>
> And for the code that calculates results across NASBenchmarks, we used their original code (almost exactly) to avoid any implementation difference. As a result, we had to make slightly different environments for the different repositories that we relied on which can not easily be merged into a single repository. Our code for metric calculation is an easy plugin function, illustrated in the ZeroCostProxy/ folder, for computing zero-cost metrics for a Pytorch network.

---

> > ### Author Response · Authors · 2023-05-02
> > **Futher Comments**
> >
> > We kindly notify the reviewer that we provide two extra repositories in anonymous  GitLab that provide updated reproduction on ZenNAS search space and a toolkit for calculating our metrics in the PyTorch network.
> >
> > 1. ZenNAS: https://gitlab.com/JuniperSling/zen-test
> >
> > 2. Toolkit: https://gitlab.com/JuniperSling/zc-toolkit
> >
> > We are happy to assist if the reviewer has any further questions.

---

> > > ### Author Response · Authors · 2023-05-04
> > > **Comments on updated source code**
> > >
> > > Dear Reviewer:
> > > We hope our updated source code can solve all the issues you mentioned, and we are more than happy to support you with any reproducibility issues you might encounter. We seek your valuable comments and believe they would help us better deliver our contributions to the NAS community.

---

### Official Review · Reviewer_QQVT · 2023-04-13

**Potential Impact On The Field Of Automl Rating:** 4
**Technical Quality And Correctness Rating:** 2
**Clarity Rating:** 3

**Summary Of Contributions:**

The authors of this paper claim two main contributions, as follows:
- revisiting and unifying the understanding of metrics used to predict architecture performance in neural architecture search (NAS)
- proposing novel metrics that outperform existing ones from a theoretical and practical perspective

**Actions Required To Increase Overall Recommendation:**

- a thorough experimental assessment of the metrics considered and proposed
- a revision of the paper, making the theoretical discussion more straightforward and substantiating the paper with experimental results

**Clarity:**

The paper is generally well-written, though its structure and style of writing differ from the traditional NAS literature. The most significant drawback, however, lies in the brevity of the assessment part of the paper. Given that the theoretical analysis section was rather long and verbose, it would likely be better to have part of it provided as supplementary material so the assessment could be further developed.

**Overall Review:**

This paper revisits and unifies the understanding of metrics used to predict architecture performance in neural architecture search (NAS). In addition, authors also propose novel metrics that they claim outperform existing ones from a theoretical and practical perspective. The topic is relevant and, if contributions were achieved, the paper would have significant impact on the field, since it brings a better and broader understanding of existing techniques. In addition, given the relevant of metrics to predict architecture performance in NAS, improved metrics are very welcome and would be readily adopted by the community. Furthermore, the paper is generally well-written, though the assessment part of the paper is considerably brief. Since the theoretical analysis section was rather long and verbose, there is clearly room to better balance the different parts of the paper. More importantly, the experimental assessment is very limited for contributions to be validated. Nonetheless, since the theoretical part of the paper escapes my area of expertise, I evaluate it as a borderline paper.

**Potential Impact On The Field Of Automl:**

If contributions were achieved, the paper would have significant impact on the field for at least two major reasons:
- given the speed with which novel NAS approaches are proposed, any paper proposing to better structure the literature is welcome, especially when it brings a better and broader understanding of existing techniques
- given the relevance of metrics to predict architecture performance in NAS, improved metrics are very welcome and would be readily adopted by the community

**Review Confidence:**

2: You are willing to defend your assessment, but it is quite likely that you did not understand the central parts of the submission or that you are unfamiliar with some pieces of related work.

**Review Rating:**

6: Borderline Leaning Accept: Technically sound paper where reasons to accept outweigh reasons to reject. Please use sparingly.

**Review Summary:**

This paper addresses metrics used to predict architecture performance in neural architecture search (NAS). In detail, authors revisit, unify, and propose novel metrics. The topic is relevant, since it brings a better and broader understanding of existing techniques. Furthermore, the paper is generally well-written, though the assessment part of the paper is considerably brief. More importantly, the experimental assessment is very limited for contributions to be validated. Since the theoretical part of the paper escapes my area of expertise, I evaluate it as a borderline paper.

**Technical Quality And Correctness:**

The theoretical analysis presented in the paper escapes my expertise, but seems logically sound. However, this work revisits previous theoretical works precisely because experimental analysis can reveal behavior that indicate that the proposed theory is either incomplete or at least immature. In this context, the paper is very welcome, but the experimental assessment is very limited for contributions to be validated.

---

> ### Author Response · Authors · 2023-05-01
> **Thank you for your feedback.**
>
> We thank the reviewer for the positive feedback on our work. We hope to have answered all of your questions satisfactorily below; Please do let us know if you see any further issues in the paper that need to be clarified or addressed.
>
> > The Theoretical analysis section is too long, while the experimental lack sufficient analysis.
>
> We thank the reviewer for the constructive feedback on our paper. We will make the theoretical part more concise and provide further analysis of our results to justify our contribution in the camera-ready version. Please also refer to our other response for further experiments that clarify the contribution of our approaches.

---

### Official Review · Reviewer_7dfr · 2023-04-13

**Potential Impact On The Field Of Automl Rating:** 4
**Technical Quality And Correctness Rating:** 4
**Clarity Rating:** 3

**Summary Of Contributions:**

The authors provide a deeper understanding of various zero-cost proxies by interpreting them based on a pruning perspective, and they propose a new combination of zero-cost proxies which achieve state-of-the-art performance. More specifically, they introduce a new gradient-based metric called SSNR and use layer-wise naswot as an activation-based metric to propose a combined proxy called T-CET. And the superiority of the proposed proxy is confirmed through various experiments.

**Actions Required To Increase Overall Recommendation:**

It would be better if more information about the experimental setup could be added. Additionally, although it seems to be due to the page limit, the size of Table 2 is too small to be easily read on the typical A4 size.

**Clarity:**

The manuscript is well-written and easy to understand in most parts. It would be better if the details about the experimental setup is included in Section 4.

**Overall Review:**

The strengths of this paper lie in the analysis of various zero-cost proxies in the pruning literature, the proposal of a new proxy that combines existing proxies, and the demonstration of high performance. The theoretical and experimental analysis of the paper is expected to be of great help to future research in the zero-cost NAS field. However, the lack of detailed descriptions of the experimental environment and the brief analysis of the experimental results serve as drawbacks.

**Potential Impact On The Field Of Automl:**

Zero-cost NAS is a relatively recent and actively researched technique among NAS variants. This paper not only provides a deeper understanding of the zero-cost proxies used in NAS but also proposes a new combined proxy that exhibits better performance than existing proxies. Although the proposed method couldn't beat existing methods in all search spaces and datasets, theoretical analysis and experimental analysis on various zero-cost proxies will be very helpful in the related field. In my opinion, this paper demonstrates the significant contribution to the zero-cost NAS field.

**Review Confidence:**

3: You are fairly confident in your assessment. It is possible that you did not understand some parts of the submission or that you are unfamiliar with some pieces of related work.

**Review Rating:**

9: Strong Accept: Technically flawless paper with major impact and strong evaluation, with no obvious flaws. Should be highlighted in the program.

**Review Summary:**

This paper is technically correct and the proposed method consistently beat comparing methods on various search spaces and datasets. Also, the authors provide a novel view to analyze zero-cost proxies. Therefore, my recommendation is a strong accept.

**Technical Quality And Correctness:**

The theoretical analysis and the process of deriving the proposed proxy in this paper are technically sound.

---

> ### Author Response · Authors · 2023-05-01
> **Thank you for your support of our work**
>
> We thank the reviewer for the positive feedback, and appreciate the time and effort that the reviewer has shown to help us improve our work. We have further updated our paper and hope to have incorporated all your suggestions satisfactorily below.
>
> >  The lack of a detailed description of the experimental environment and a brief analysis of the experimental results.
>
> We thank the reviewer for the constructive feedback on our paper. We will make the theoretical part more concise and provide further analysis of our results to justify our contribution for the camera-ready version.
>
> > … The font size of Table 2 is too small to read
>
> We thank the reviewer for raising this concern and have accordingly increased the size of table 2 by around 20% for the camera-ready version.